# Design and Implementation of *SEMAR* IoT Server Platform with Applications

**DOI:** 10.3390/s22176436

**Published:** 2022-08-26

**Authors:** Yohanes Yohanie Fridelin Panduman, Nobuo Funabiki, Pradini Puspitaningayu, Minoru Kuribayashi, Sritrusta Sukaridhoto, Wen-Chung Kao

**Affiliations:** 1Graduate School of Natural Science and Technology, Okayama University, Okayama 700-8530, Japan; 2Department of Informatic and Computer, Politeknik Elektronika Negeri Surabaya, Surabaya 60111, Indonesia; 3Department of Electrical Engineering, National Taiwan Normal University, Taipei 106, Taiwan

**Keywords:** *Internet of Things*, server platform, *SEMAR*, IoT application system, sensor, MQTT, *REST API*

## Abstract

Nowadays, rapid developments of *Internet of Things (IoT)* technologies have increased possibilities of realizing *smart cities* where collaborations and integrations of various *IoT application systems* are essential. However, IoT application systems have often been designed and deployed independently without considering the standards of devices, logics, and data communications. In this paper, we present the design and implementation of the *IoT server platform* called *Smart Environmental Monitoring and Analytical in Real-Time (SEMAR)* for integrating IoT application systems using standards. *SEMAR* offers *Big Data* environments with *built-in* functions for data aggregations, synchronizations, and classifications with *machine learning*. Moreover, *plug-in* functions can be easily implemented. Data from devices for different sensors can be accepted directly and through network connections, which will be used in real-time for user interfaces, text files, and access to other systems through *Representational State Transfer Application Programming Interface (REST API)* services. For evaluations of *SEMAR*, we implemented the platform and integrated five IoT application systems, namely, the *air-conditioning guidance system*, the *fingerprint-based indoor localization system*, the *water quality monitoring system*, the *environment monitoring system*, and the *air quality monitoring system*. When compared with existing research on IoT platforms, the proposed *SEMAR* IoT application server platform offers higher flexibility and interoperability with the functions for IoT device managements, data communications, decision making, synchronizations, and filters that can be easily integrated with external programs or IoT applications without changing the codes. The results confirm the effectiveness and efficiency of the proposal.

## 1. Introduction

The rapid growth of urban populations has increased the risk toward *quality of life (QoL)* around the world [1]. *Smart cities* have been studied for identifying, preventing, and acting in certain situations. In smart cities, *QoL* is commonly handled with indicators that measure the effectiveness of the services and sustainability of a city in domains/verticals, such as environments, health cares, securities, transportation, economies, educations, and governments [2]. Particularly, the environment vertical has drawn special attention in recent years. Indicators of environmental pollutants, such as air and water quality, road conditions, and house conditions, must be monitored to detect adverse situations associated with overpopulated regions. In this sense, *Internet of Things (IoT)* applications must provide interoperability tools that collect, store, and disseminate data from several sensors, and provide them to other systems [3,4]. Thus, *smart cities* require collaboration and integration of various IoT application systems. Studies of IoT server platforms have emerged for such purposes, where several challenges hinder better management and analysis of IoT application data using platforms.

The first challenge involves the lack of a common data format between data sensors and data communication protocols. For instance, to measure the air and water quality, different sensors with different geo-location concepts such as addresses, buildings, regions, or cities can be handled in different ways. Thus, an IoT server platform should be able to handle various data types from different sensors, which makes it necessary to be able to work with each other despite the diversity in communication protocols or sensor technologies.

The second challenge is the standard parameters for data processing. As an example, the majority of air quality monitoring systems use *Air Pollution Index (API)* to define the indicators of the carbon monoxide (CO), the nitrogen dioxide ($NO_{2}$), the sulfur dioxide ($SO_{2}$), the ozone ($O_{3}$), and the particulate matter ($PM_{10}$) [5]. However, other researchers mentioned that it might be necessary to consider other indicators such as the temperature and the humidity for their measurements [6].

The third challenge concerns the data interoperability between various IoT application systems within the same domain. It can be described as the integration of plural systems by sharing output data through information networks [3]. For example, a smart building system should integrate the human *Indoor Positioning System (IPS)* with the environment monitoring system to improve *QoL* while reducing energy usage.

However, in general, IoT application systems for smart cities have been designed without considering these challenges. They have been deployed independently and cannot be integrated with other systems.

In this paper, we propose an IoT server platform called *Smart Environmental Monitoring and Analytical in Real-Time (SEMAR)* for integrating various IoT application systems. *SEMAR* is able to offer *Big Data* environments with rich *built-in* functions for data aggregations, synchronizations, and classifications with *machine learning*. Moreover, *plug-in* functions can be easily implemented and added there. Data from devices for different sensors can be accepted directly and through network connections, which will be used in real-time for user interfaces, text files, and access to other systems through *Representational State Transfer Application Programming Interface (REST API)* services.

For evaluations of *SEMAR*, we implemented the platform and integrated five IoT applications, namely, the *air-conditioning guidance system*, the *fingerprint-based indoor localization system*, the *water quality monitoring system*, the *environment monitoring system*, and the *air quality monitoring system*. The results confirm the effectiveness and efficiency of the proposal, including the reduction in the data transmission delay with the implemented *Message Queue Telemetry Transport (MQTT)* service [7].

The rest of this paper is organized as follows: Section 2 presents related works. Section 3 presents the design of *SEMAR*. Section 4 presents the implementation of *SEMAR*. Section 5, Section 6, Section 7, Section 8 and Section 9 briefly describe the IoT application systems implemented in the *SEMAR*. Section 10 describes comprehensive performance evaluations and comparative analysis with similar related work. Section 11 presents the threats to validity. Finally, Section 12 concludes this paper with future works.

## 2. Related Works

In [8], Kamienski et al. proposed a three-layered Open IoT ecosystem approach for smart application architectures. It includes *input*, *process*, and *output* in IoT application systems. The *input* gathers information from multiple sources, such as sensors and other services. The standard communication protocols cover the device connections. The *process* is given by a collection of methodologies, procedures, and algorithms for effective and efficient data processing. The *output* provides capabilities for data visualizations and accessibility.

In [9], Bansal et al. proposed five layers for the IoT application system architecture, consisting of perception, transport, processing, middleware, and application. They divided data processing into two layers, where the processing layer concentrates on filtering and formatting the data, and the middleware layer intends to execute various logical and analytic operations.

The connectivity in IoT systems was discussed in [10], where Li et al. examined the networking technologies, and described that IEEE 802.11 (WLAN), IEEE 802.15.1 (Bluetooth, Low-energy Bluetooth), IEEE 802.15.6 (wireless body area networks), and 3G/4G were the most widely adopted in IoT application systems and smart city environment systems.

Since IoT systems might consist of various physical things and sensors, it is essential to provide a device-to-device communication protocol. Malche et al. in [11] proposed the *MQTT* communication protocol for environmental monitoring systems with multiple device sensors. Sharma et al. in [12] defined the *Representational State Transfer (REST)* web service as the gateway to collect device data through the *HTTP POST* protocol. Zhang et al. in [13] studied *NATS* open source messaging to communicate between IoT devices [14].

The numerous options of communication protocols were surveyed by Dizdarevic et al. in [15]. They concluded that *MQTT* and *HTTP POST* are the most suitable for IoT application systems since they are well matured and stable.

In [16], Marques et al. proposed the IoT system architecture for the *indoor air quality (IAQ)* system in a laboratory environment named *iAQ Plus (iAQ+)*. It collects data from devices through Wi-Fi connections and stores data in the SQL server. The authors proposed a web portal and mobile application to manage and visualize the obtained data; however, the system does not offer data analysis functions to process sensor data.

In [17], Benammar et al. proposed the *IAQ* system that is integrated with the *Emoncms* IoT platform for storing and visualizing air quality data, temperature, and humidity. The authors used a *Waspmote* microcontroller connected to Raspberry PI as sensor nodes and the *MQTT* service to send data.

In [18], Mandava et al. proposed to integrate machine learning algorithms and the IoT platform infrastructure for monitoring air pollution in smart cities. The system collects environmental and location data to determine air pollution conditions in specific areas, and uses the collected data to build a data model for air pollution detections using supervised machine learning algorithms. The experiment results confirmed the effectiveness of the proposed data model for air pollution detections.

In [19], Senožetnik et al. proposed a management framework for groundwater data in smart cities. The system uses a web-based IoT service to receive data through *HTTP POST*, convert it into the *JavaScript Object Notation (JSON)* format, and store it in the *MongoDB NoSQL* database. It also allows sharing collected data through *REST API*. This system is similar to our proposed one; however, the system only provided data communications through *HTTP POST*. Moreover, it did not implement data processing functions to analyze the obtained data.

In [20], Kazmi et al. proposed a platform that provides interoperability of diverse IoT applications in smart cities named *VITAL-OS*. It can be integrated with other IoT application systems through *REST API*.

In [21], Toma et al. proposed an IoT platform for monitoring air pollution in smart cities. The system contains wireless and wired connections with sensors to send data through *MQTT* communications to the server using cellular networks. It allows sharing data through *REST API*; however, this platform was built and implemented only for this IoT application of monitoring air pollution.

In [22], Javed et al. proposed an IoT platform for smart buildings. It consists of the discovery, storage, and service planes. The discovery plane performs connectivity control with devices through *HTTP* communications. The storage plane manages data storage using *Apache Cassandra* [23]. The service plane provides data processing composed of data indexing, visualizing, and analysis.

In [24], Badii et al. proposed an open source IoT framework architecture for smart cities called *Snap4City*. The system offers modules for device managements, data processing, data analysis, and data visualizations.

In [25], Putra et al. proposed an implementation of wireless sensor networks in smart cities to monitor air pollution. A device will transmit data regarding the air pollution to a server through a Wi-Fi network.

In [26], Gautam et al. proposed an IoT application for the water supply management system in smart cities. The proposed architecture uses *General Purpose Input Output (GPIO)* communications for connecting ultrasonic sensors and water pumps with Raspberry PI. Moreover, it uses Ethernet cables as network interfaces to Raspberry PI and the router. It offers data analysis services and real-time predictions using machine learning algorithms for processing data; however, this framework is only built for this single IoT application.

In [27], Oliveira et al. proposed an IoT application for road environment monitoring using mobile-based sensors. The system receives sensor data through *HTTP POST* communications in the JSON format, and allows processing and visualizing it. It also provides a function to export data in CSV files.

In [28], Metia et al. proposed a digital filter for the IoT-based air pollution monitoring system. The experiment results in this study show that the data processing using *Kalman* filter has enhanced the reliability and accuracy of the system; however, they did not implement the real-time data processing.

In [29], Twahirwa et al. proposed the system for monitoring roads, weathers, and environments by attaching multiple sensors to a vehicle and sending sensor data to the IoT server. The IoT server can process, store, and visualize data with the web application system.

In [30], D’Ortona et al. showed the benefits of implementing *MQTT* communications in IoT application systems for smart cities. The *MQTT* communications allow the construction of highly scalable and flexible IoT systems.

In [31], Kumar et al. proposed an *anomaly-based intrusion detection system (IDS)* to secure IoT networks from threats such as spying and malicious controls. It was implemented at the fog node level. The proposed approach might be adopted as an additional system that can avoid threats before the IoT platform receives them. Moreover, in [32,33], a method was proposed to protect IoT networks by data pre-processing functions. It comprises feature mapping, missing value inputting, normalization, and feature selection techniques. The proposed method is similar to our approach in the data aggregation function. *SEMAR* only processes and stores sensor data registered in the sensor format data storage.

In [34], Kumar et al. designed PEFL for secure open communication channels in IoT application systems. The proposed method utilized *Long Short-Term Memory (LSTM)* and privacy encoding techniques in order to reduce security risk and maintain privacy. Moreover, in [35], authors proposed an framework for preventing cyber attacks on IoT-fog computing. It offered virtualized northbound interfaces such as load balancer and resource management to manage networks in IoT systems. The proposed architecture can be utilized to enhance network performances for the *SEMAR* IoT application server platform in future works.

## 3. Design of *SEMAR* IoT Server Platform

In this section, we present the design of the *SEMAR* IoT server platform for integrating various IoT application systems.

### 3.1. System Overview

Figure 1 shows the proposed design of the *SEMAR* IoT server platform. The main components are *data input*, *data process*, and *data output*. The *data input* is responsible for accepting data from various sources. It consists of network interface devices and communication protocols. The *data process* provides the modules for data processing, control, and collection. The *data output* enables visualizations and sharing of collected data. In Table 1 we summarize the nomenclature of all the symbols and variables used in this paper.

### 3.2. Data Input

*SEMAR* needs to collect data from a number of different devices using various network connectivity and communication methods; therefore, the following network interfaces for constructing physical network connections are implemented in the platform, where standard IoT communication protocols for data transmission, namely *HTTP* and *MQTT*, are included. In the context of IoT, physical devices as a perception layer consist of a number of sensors connected to a controller. With the growth of IoT technology, controllers such as Arduino and Raspberry PI have provided diverse network connectivity to accept data from various sensors. *General Purpose Input Output (GPIO)* is the programmable interface in the device controller to receive or send command signals from/to IoT sensor devices [36]. In IoT application systems, GPIO is the standard interface for connecting sensor devices with the controller. In addition, it is used for connecting controllers with external modules such as Wi-Fi ones for data communications. *Universal Serial Bus (USB)* is the serial communication media to link devices with computers via USB ports. Currently, numerous sensor instruments and devices can transmit data using USB connections. The USB connection offers a high data transfer capacity. In addition, external communication modules such as Wi-Fi for data communication can also be added using USB connection.

Regarding the IoT data transmission concept, diverse hardware and software connectivity should be considered. Diverse network interfaces utilize hardware-based transmissions, such as Wi-Fi, Ethernet, and Cellular—this enables machine-to-machine and device-to-server communication.

*IEEE 802.11 wireless LAN (Wi-Fi)* is the most prevalent network interface in IoT systems. It connects devices with each other and to servers. Wi-Fi is useful to connect a lot of devices regardless of their locations with computers, which improves IoT application developments. Ethernet offers secure and dependable *wired* connectivity. It is one of the most used network interfaces in IoT systems; however, the implementation can be difficult over long distances.

Although Wi-Fi and Ethernet offer excellent network performance, we should consider their security and coverage area. The alternative network interface that can be utilized is *cellular* networks. *Cellular* is the network interface allowing the mobility of devices with the existing widespread availability of *cells* to connect with the internet. Currently, 5G cellular connections offer solutions with wider bandwidths than Wi-Fi or Ethernet. The IoT platform can use it through Wi-Fi interfaces with mobile routers.

The last part of *Data Input* is the communication protocol between IoT devices and servers. An IoT server should support publish–subscribe and push-and-pull messaging systems for sending and receiving data. Thus, our proposed system utilizes *MQTT* and *REST API* for the communication protocol service.

*MQTT* is one of the protocols that have been designed for data communications in IoT application systems. It can work with minimal memory and the processing power [37]. The *MQTT broker* works for receiving messages from clients, filtering the messages according to a topic, and distributing the messages to subscribers [38]. The *MQTT broker* is implemented in the IoT gateway function of the platform to provide data communication services in *SEMAR*. The IoT gateway function offers communication services to connect sensor devices to the server. Using this protocol, sensor devices can transmit messages containing sensor data in the JSON format with *MQTT* topics. By subscribing data at the same *MQTT* topic, the data aggregation program in the platform obtains each sensor data. In addition, the study by Al-Joboury in [39] shows that the load balancer can increase the performance and the capacity of *MQTT* data communications.

The IoT gateway function also implemented the *REST API* for receiving sensor data through the *HTTP POST* communication protocol. It can only receive data in the JSON format. The *REST API* provides URLs for sensor data transmission. The management function in the platform creates the unique URL for each device. The *HTTP POST* communication protocol is compatible with standard network interfaces. Using *REST API*, sensor devices can transmit data in the JSON format.

### 3.3. Data Process

The *data process* in the *SEMAR* server platform offers various functions. The large amount of data from *data input* will be processed to obtain meaningful information using some functions. The functions are implemented as independent modules to reduce system crashes at system failures. They can be extended to *microservices* [40,41]. The concept of *microservices* is the method of developing a large-scale system with a set of small independent services. For their implementations, thread-based programs are adopted to improve their performances for real-time data processing. Each service will initiate a new thread to process the newly coming data.

#### 3.3.1. Data Management (Storage, Aggregator, and *Plug-in* Functions)

The data management system is the main function of the IoT platform. In the context of IoT, systems must provide data storage, transaction management, query processing, and data access for application systems. Thus, the IoT platform must offer services to process the data flow from input to output. Moreover, towards developing diverse IoT applications, devices involved in IoT should be able to generate different kinds of data types according to the application.

In order to provide various IoT application systems, *SEMAR* should be a useful platform for a variety of IoT application systems. Thus, it needs to support massive amounts of data in different formats. Moreover, it needs to store all the necessary data by offering data storage for every application. The *management data storage* is the database that stores the operating parameters in the *SEMAR* server platform including the implemented IoT application systems. The data include the information regarding connected devices, communications, and parameters for the process modules running on the platform. On this platform, each device has its own unique sensor format. The management data storage database keeps the sensor format as the template to help the development of an IoT application system on this platform.

Meanwhile, the *sensor data storage* is the database that stores all the sensor data in the platform. In IoT application systems, sensor devices may offer various data and it may change it over time with unstructured formats; therefore, the platform uses the *big data* technology to store unstructured JSON objects generates the unique data storage for each device. This data storage utilized only accepts registered device data; therefore, we implement additional data stored in the form of Log files. *Log Files* are used to keep the values of any defined or undefined data using the CSV format. The defined data represents a sensor data that fits the format registered in the management data storage. The undefined data represents data whose format is not registered.

The *schema data storage* is the database that can be used to help the users of IoT application systems by dynamically specify the names, fields, and data types in accessing this storage. It supports multiple data types, including integer, float, date, time, date-time, and string.

Figure 1 illustrates that this database is used to store data synchronization results. Through REST API, other systems can access to the sensor data storage. As the advantage of this database, it can be dynamically defined and modified by the user. It can assist integration of various complex IoT application systems.

In an IoT system, the data lifecycle begins with the communication gateway receiving sensor data, continues with data aggregation and preprocessing, and concludes with data storage. For this purpose, SEMAR provides a *data aggregator* function. The *data aggregator* is the module of collecting data from various data sources, applying the value-added processing, and repackaging the information in a consumable format. Algorithm 1 illustrates the data processing procedure in this module. It forwards the result to the following data filter or stores it in the data storage through the database access.

The data management system plays a role in the sensor data storage process and provides access to additional data processing functions. Those services are not only for systems integrated in the IoT platform (*built-in*) but also for *plug-in* functions that may be deployed as an extension. Because an IoT application system may require unique data processing that has not been implemented in the platform. Thus, the platform is designed and implemented so that users can easily implement *plug-in* functions without modifying existing codes, to fulfill the demands of IoT application systems. The *plug-in* functions can use *REST API* to access the data in the platform.
**Algorithm 1** Data aggregator
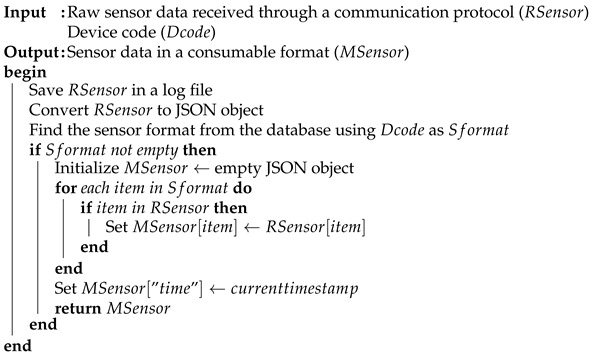


#### 3.3.2. Data Filter and Synchronization

In this research, we additionally explore the data processing capabilities required by IoT applications that are not included in the standard data management services. For example, sensors of IoT devices may generate measurement errors and noise during the measuring process. It can impact the risk of data analysis problems. In addition, IoT applications such as indoor localization systems require real-time sensor data from several devices simultaneously; therefore, our platform deploys the data filter and synchronization functionalities for processing sensor data.

The functions of filtering sensor data before being saved in a data storage are implemented. Digital filters are adopted to reduce noise and inaccuracies in data. The following procedure is applied for filtering data:It receives sensor data in a JSON format.It selects the sensor field’s value to be filtered.It add the field value in the JSON object with the filter result.It stores the JSON object in the database.

The data synchronization function can synchronize the data from different devices by referring to the timestamp in the data store schema. The *timestamp* was given when the platform receives the data from the sensor device. Thus, the platform requests the data from each sensor’s storage at a specified detection time. For each sensor data, the field for the identifier (Fi) to group sample data in a specific value, the field for the value (Fv) to be synchronized, the default value (default), and the four functions to process the data are prepared. The following functions are implemented to process the data:*Average*: it returns the average value of the data collected during the detection time.*Current*: it returns the last value among the data collected during the detection time.*Max*: it returns the highest value among the data collected during the detection time.*Min*: it returns the lowest value among the data collected during the detection time.

Algorithm 2 illustrates the data processing procedure in this module.
**Algorithm 2** Data Synchronization
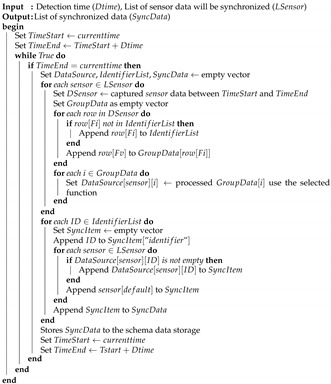


#### 3.3.3. Machine Learning and Real-time Classification

One of the exploitation scenarios for the massive quantity of IoT data is its predictive capability by utilizing machine learning approaches. Several researchers approved the effectiveness of machine learning implementation in IoT applications [42,43,44]. Moreover, Kumar et al. in [45] proposed the ensemble design combining machine learning algorithms to protect networks on the Internet of Medical Things in real-time; therefore, we implement machine learning and real-time classification function in *SEMAR*.

The machine learning algorithms are implemented to help data classifications. The *Support Vector Machine* [46,47] and *Decision Tree* [48,49,50] are implemented in this platform as standard machine learning algorithms in IoT application systems.

*Decision Tree* employs tree decisions including event outcomes, resource costs, and utility costs. It can create a data model for predicting outcomes by learning simple decision rules according to the data features. The data model structure consists of internal nodes representing an attribute, branches representing a decision rule, and leaf nodes indicating an outcome. Here, *C4.5*, *CART (Classification and Regression Trees)*, and *Naive Bayes Tree* are selected and incorporated into the platform as the most well-known machine learning algorithms [48]. *CART* is the binary recursive partitioning method that can handle both numerical and category data [48,49,50]. It can determine the impurity degree of acceptable data and build a binary tree in which each internal node provides two classes for the accepted attribute. The tree is formed by iteratively picking the attribute with the lowest *Gini* index. The *Gini* index for each node is calculated by the following equation [48]:(1)$$Gini\left(t\right)=1-\sum_{i=1}^{n}P{\left(i|t\right)}^{2}$$

*Support Vector Machine (SVM)* is utilized as the regression and classification technique [51]. This approach has been used for the big data classification [47]. The SVM computes linear decision boundary lines that can separate the data for the labeled groups. The SVM decision boundary line is calculated by the following equation:(2)$$f\left(x\right)=\sum_{\forall i}^{}y_{i}\alpha_{i}K\left(x_{i},x\right)$$
where $y_{i}$ represents the class label, $\alpha_{i}$ represents the learned weight, K() represents the kernel function, $x_{i}$ denotes the support vector, and *x* denotes the labeled training sample data. The kernel function is given by a collection of mathematical operations used to process the input data and convert it into the required format. The *radial basis function (RBF)* kernel is one of the common kernel functions in SVM. The following equation illustrates the formula of the (RBF) kernel:(3)$$K\left(x_{i},x_{j}\right)=\exp\left(-\frac{d{\left(x_{i},x_{j}\right)}^{2}}{2l^{2}}\right)$$
where *l* represents the length scale of the kernel and $d(x_{i},x_{j})$ denotes the Euclidean distance between $x_{i}$ and $x_{j}$.

*Decision Tree* and *SVM* have several hyper parameters. For them, the *Randomized Search Method* is implemented in *SEMAR* to find the optimal combination of hyper parameters, due to its superior performances with the low cost and short computing time compared to other methods.

For reference, the *Decision Tree* algorithm has the following hyper parameters:*Maximum depth (max_depth)*: represents the maximum depth of the tree model result. It is used to select the optimal model to prevent over-fitting.*Minimum samples split (min_samples_split)*: represents the minimal amount of data required to separate an internal node. If it is large, it can prevent over-fitting; however, if it is very large, it can cause under-fitting.*Minimum samples leaf (min_samples_leaf)*: represents the minimal amount of data required to be left at the leaf node. It is similar to the *minimum samples split* parameter.*Minimum weighted fraction leaf (min_weight_fraction_leaf)*: represents the total weight required at a leaf node.

The *Support Vector Machine* algorithm has the following main hyper parameters:*Kernel*: represents the function of transforming the training dataset into the higher dimension space. The standard kernel consists of *Radial Basis Function (RBF)*, *linear*, *polynomials*, and *sigmoid*.*C*: represents the penalty parameter that controls the trade-off between the decision boundary and the misclassification. *C* value controls the margin of the decision boundary line to avoid misclassifications. The large value can prevent the model from allowing any misclassification. If the dataset is linearly separable, it will work; however, if the dataset is non-separable/nonlinear, it is better to use a small *C* value to avoid overfitting, although it allows misclassifications to occur.*Gamma*: represents the coefficient of the kernel used to decide the curvature of the decision boundary line. The value of *Gamma* determines the shape of the decision boundary line according to the number of dataset points. The large value causes the decision boundary to be easily affected by fewer data points, and the shape becomes complex. It can be helpful for nonlinear datasets; however, if it is too large, it tends to be over-fitting. On the other hand, for the linear dataset, the small values make the decision boundary line more general and useful.

The machine learning algorithms allow the user to use the data stored in the data storage as the sample data. This module can generate a data model for the real-time data classification module.

The real-time data classification function is implemented to analyze a huge amount of data from various sensor devices by periodically running the following procedure:1.It loads the data classification model made by the machine learning algorithm.2.It receives sensor data from the database.3.It classifies data into classes by running the data model.4.It stores results in the database.

The classification model can be created by each user separately. Moreover, the user can start or stop the real-time data classification at the user interface.

### 3.4. Data Output

Several output components, such as the monitor display, the user interface, the data export, REST API, and the notification function, are considered to use the data in the platform. The monitor display is attached to the sensor node, and accesses the user interface in the platform through a network connection. It can easily show sensor data for each device.

The user interface is provided at the web browser to allow users to see the sensor and synchronized data by tables, graphs, or maps. The platform allows users to access the sensor data using the time of data receipt. It receives the sensor data in the JSON format by accessing *REST API*. The column in the table is formed automatically based on the sensor format of each device. The platform can generate the graph for each registered format sensor. Visualization maps will display the data in digital maps based on the GPS data. The data export feature is designed and implemented to allow users to download data in Excel, JSON, text, or CSV format at the specified time. Users can use this feature by accessing to the user interfaces.

*REST API* is employed as a back-end system to access the sensor data. The sensor data are retrieved from the database and is converted to the JSON format. It will be sent to the user interface and *plug-in* functions using *HTTP POST* communications. The platform can exchange and integrate data with other IoT application systems via *REST API*.

The notification function allows the user to define the threshold for each sensor data as the trigger of the message notification. If the value is over the threshold, the platform will send a notification. The platform offers two different communication services. First, it publishes a message to a specific topic using the *MQTT* communication protocol. Thus, the IoT application system can subscribe to topic to receive the messages. Second, it delivers email notifications through the mail server service installed on the server platform. The user can dynamically define email recipients.

### 3.5. Management Service

The *management service* is used to manage all functions in the *SEMAR* platform. It includes the managements of users, devices, communications, schema data, synchronization functions, analytics, data filters, and notification functions. The management of users allows us to add users, set permissions, and restrict access to the devices.

The device management service provides the functions to register the devices and the sensors of the IoT application system. It allows managing the sensor format for each device dynamically. The platform can process, save, and display the data registered in the sensor format. For convenience, the *SEMAR* platform provides a template to add the device with the same sensor format easily. The schema data management allows to create the schema database, define the field format, and manage the data.

The management service provides the functions to add, update, and delete settings for data synchronizations, data analytics, data filtering, and notifications. It allows the user to run and terminate the module service in the data process. All the configuration settings are saved as JSON objects.

## 4. Implementation of *SEMAR* IoT Server Platform

In this section, we present the implementation of the *SEMAR* IoT server platform. Table 2 shows the summary of the implementation.

In this implementation, the following two types of communication protocol services are implemented for data input. *Mosquitto* [52] is installed for the *MQTT* broker. It allows the platform to receive messages through various *MQTT* versions, and supports connections from Wi-Fi, Ethernet, and Cellular network interfaces. Then, *REST API* is implemented based on Python programming and *Tornado* web server [53]. It allows the platform to receive messages through *HTTP POST* and supports connections from Wi-Fi, Ethernet, and Cellular network interfaces.

The data process is deployed and implemented in the platform. They are developed in Python using a variety of modules and dependencies. For IoT data management systems, we used two different databases service implemented in the platform according to the design in Section 3. The Big Data repository *MongoDB* [54] is utilized for the data storage for managements, sensors, and schema. *MongoDB* saves data in the JSON format as the flexible approach—there is no need to define data structures, unlike *SQL*. In addition, the log file is implemented in the CSV format. It can be accessed using a file controller library in Python.

Two different data aggregators are implemented. The first one enables message receptions using the *MQTT* communication protocol. It allows a different *MQTT* communication settings for each sensor device. The second one does it with *REST API*. Both data aggregators access the data storage via *PyMongo*.

In this study, the data filter and synchronization capabilities are utilized to process sensor data. *Scipy* and *KalmanFilter* Python libraries are used to apply the data filters. After filtering the data, *PyMongo* is used to save it in the data storage. The data synchronization used *PyMongo* for sensor data in the data storage. *Pandas* is used for grouping data sensors. *Threading* library is used to enhance the performance of the platform. This function runs periodically on the server based on the detection time. The user can stop and start this service at the administration page in the user interface. Figure 2 illustrates the user interface of the data synchronization function for the sensor data during 30 s.

According to the design systems in Section 3, the data analysis systems consist learning process and real-time analysis service. We implemented both services in Python. *Scikit-learn* [55] is used to facilitate the learning process. The *Sklearn library* is utilized for real-time analysis to make the classification model during the learning process.

Data output includes the data visualization and the data sharing with other systems including the *plug-in* systems. The *CodeIgniter* PHP Framework is adopted to create user interfaces based on the *Model-View-Control (MVC)* design paradigm [56]. A user interface will offer data visualizations using *HighchartJS*, *DataTalbes*, and *OpenStreetMap*. Here, *Apache* and *PHP* are required. Figure 3 shows the table of sensor data. Figure 4 show graphs of sensor data.

*DataTables* library is used to allow the user to download sensor data in Excel, JSON, text, and CSV formats at the specified times. Figure 5 show the data export interface. *REST API* is built with Python and *Tornado*. It allows other application systems and *plug-in* functions to access to sensor data in JSON formats.

Finally, The management service is built in Python and *Tornado* web server. It allows us to receive messages through *HTTP POST*, and to access data storage by *PyMongo*.

## 5. Integration of Air Quality Monitoring System

As the first IoT application system, the *air quality monitoring system* is integrated in the proposed platform. It can monitor the air quality in smart cities.

### 5.1. System Architecture

Figure 6 shows the system overview. This system uses a single-board computer (SBC) that is connected to the GPS sensor device and the air quality sensor device through Wi-Fi. The air quality sensor device covers the carbon monoxide sensor (MQ7), the particulate matter sensor (Shinyei PPD42), the sulfur dioxide sensor (MQ135), the ozone sensor (MQ131), and the nitrogen dioxide sensor (MiCS 2714. The sensor sends the voltage measurement data to the *Arduino UNO* via GPIO. *Arduino UNO* converts the data into the value of the pollutant concentration level and sends it to the SBC via the *MQTT protocol*. When the air sensor data are received, the SBC adds the current time and the location information (latitude and longitude) from the GPS sensor to the air sensor data, and sends it in the JSON format every five seconds through the MQTT connection.

### 5.2. Implementation in Platform

Figure 7 shows the flow of the functions in the *SEMAR* server platform for integrating this IoT application system. Through the *MQTT* connection, the data aggregator receives the sensor data and stores it in the data storage. The real-time classification estimates the air quality index from the data between 0 and 4 that corresponds to the air quality categories of good, moderate, poor, very poor, and hazardous. The output data are shown at the user interface.

To evaluate the system integration, we run the system to monitor actual air quality conditions. The sensor device is mounted on the vehicle, and the single-board computer system is placed inside the vehicle during the experiment. The device system sends air quality and GPS data every five seconds. The evaluation results show that *SEMAR* has successfully received the sensor data, processed it, and classified the air quality index based on it. The results can be displayed on the user interface in real-time. Table 3 shows the evaluation results of the classification model used in this experiment. We compare two algorithms consisting of *Support Vector Machine (SVM)* and *Decision Tree (DT)*.

Table 3 illustrates that the accuracy of the developed model is higher than 90%; therefore, we can conclude that the real-time classification function to determine the air quality in *SEMAR* provides advantages over similar studies, including the study by Toma et al. in [21].

Moreover, we also conducted experiment for hyper parameters tuning. Table 4 shows the experiment setup for optimizing the hyper parameters using the *randomized search method* in this IoT Application.

Figure 8 shows the confusion matrices for the *Decision Tree* algorithm and the *Support Vector Machine* algorithm in the air quality monitoring application. For *Decision Tree*, max_depth=12, min_samples_split=4, min_samples_leaf=9, and min_weight_fraction_leaf=0.0 are obtained, where the accuracy of the model is 99%. For *Support Vector Machine*, kernel=“linear”, C=1, and gamma=0.01 are obtained, where accuracy of the model is 95%.

## 6. Integration of Water Quality Monitoring System

As the second IoT application system, the *water quality monitoring system* is integrated. It can monitor the water quality in rivers flowing in smart cities.

### 6.1. System Architecture

Figure 9 shows the overview of the system architecture. This system utilizes the sensor device equipped with water quality sensors for the hydrogen potential (pH), the oxidation reduction potential (ORP), the dissolved oxygen (DO), the electrical conductivity (EC), the temperature, total dissolved solids (TDS), the salinity (Sal), and the specific gravity (SG). The edge computing device *Raspberry Pi 3* collects the sensor data every five seconds and sends it to a server. The system was tested at various points in the river in Surabaya, Indonesia. The sensor node detects multiple parameters of water quality.

### 6.2. Implementation in Platform

Figure 10 shows the flow of the functions in the platform for integrating this IoT application system. Through the MQTT connection, the data aggregator receives sensor data from the devices and stores it in the data storage. The real-time classification function estimates the water quality index from the collected data with a number between 0 and 3 corresponding to lightly polluted, heavy polluted, and polluted. The output data are shown in the user interface.

We evaluated the efficacy of the integration of *SEMAR* with the water quality monitoring system. The evaluation was conducted by operating the system in a real-world environment to monitor the water quality of a river. The device transmits the water sensor data to the *SEMAR* server every five seconds through *MQTT* communications. The experiment results indicate that the server received the sensor data, classified the water quality index based on the obtained data, and displayed it on the user interface in real-time. In addition, we compared the *SVM* and *DT* machine learning algorithms. are presented in Table 5 shows the evaluation results of the classification model utilized in the real-classification function.

Table 5 shows that the accuracy of the classification model for the water quality is higher than 90%. Thus, the superiority of *SEMAR* on the integration with water quality measurement systems was confirmed with abilities to receive and classify data in real-time.

## 7. Integration of Road Condition Monitoring System

As the third IoT application system, the *road condition monitoring system* is integrated. It can monitor road surface conditions in smart cities.

### 7.1. System Architecture

Figure 11 shows the system architecture overview. This system is implemented as a mobile-based sensor network attached to the vehicle. This concept is called *Vehicle as a Mobile Sensor Network (VaaMSN)*. This system consists of the edge computing device, the portable wireless camera, and the sensor device. The camera records the road conditions in front of the vehicle and transmits the image frames through *Real-Time Streaming Protocol (RTSP)*. The sensor device collects GPS, accelerometer, and gyroscopes data, and transmits them to the edge computing device via *MQTT protocol*.

The edge computing device detects potholes from the camera images using the deep learning approach, OpenCV [57], and Tensorflow [58]. When detecting a pothole, image data are recorded in the directory file. Figure 12 shows the detected pothole example by the system. The edge computing will send the location, the accelerometer, the gyroscopes, and the pothole state to the server through the MQTT connection.

### 7.2. Implementation in Platform

Figure 13 shows the flow of the functions in the platform for integrating this IoT application system. The data aggregator receives sensor data from the device through the *MQTT* connection, and stores it in the data storage. The output data appear in the user interface.

To evaluate the system integration, we run the road condition monitoring system to monitor road surfaces in actual conditions. We place the sensor device in the vehicle according to the layout shown in the system overview. They send JSON data consisting of the GPS location, accelerometer, gyroscope, and pothole status to the server through *MQTT* communications when the system detects a pothole, as shown in Figure 12. The experiment results show that the system can receive data from the device, process it, and display it on the map of the user interface in real-time.

## 8. Integration of Air-conditioning Guidance System

As the fourth IoT application system, the *air-conditioning guidance system (AC-Guide)* is integrated. It can offer the guidance for the optimal use of air-conditioning (AC) in smart cities [59].

### 8.1. System Architecture

Figure 14 illustrates the system architecture overview. *AC-Guide* uses a web camera, a *DHT22* sensor, and *Raspberry Pi 3 model b+* as the sensor device. The Python program of the system periodically (1) collects the humidity and temperature of the room and the AC control panel photo, (2) collects the standard outdoor weather data by accessing to *OpenWeatherMap API* [60], (3) calculates the indoor *discomfort index (DI)* to determines whether the indoor state is *comfort* or *discomfort*, (4) calculates the *outdoor DI* to determines whether the outdoor state is *comfort* or *discomfort*, (5) detects the on/off state of the AC from the photo, (6) sends the message to *turn on* or *turn off* the AC considering the indoor DI, the outdoor DI, and the on/off state of AC, (7) saves the data in the log file, and (8) send the data to the server using the *MQTT* connection.

### 8.2. Implementation in Platform

Figure 15 shows the flow of the functions in the platform for integrating this IoT application system.

We evaluated the effectiveness of the integration of *SEMAR* with the air-conditioning guidance system. The experiment was carried out by running the system at the #2 Engineering Building in Okayama University. The device sends JSON data containing the indoor humidity, indoor temperature, indoor discomfort index (DI), outdoor humidity, outdoor temperature, outdoor discomfort index (DI), and state of AC using *MQTT* communications every one minute. The evaluation results show that *SEMAR* can receive sensor data and display sensor data in real-time on the user interface. Previously, these data were not accessible from other systems. By integrating *SEMAR*, they can access the data through *REST API*. In addition, *SEMAR* allows adding new sensors to the system without changing the codes; therefore, the advantages of integrating the *SEMAR* system is confirmed.

## 9. Integration of Fingerprint-based Indoor Localization System

As the last IoT application system, the *fingerprint-based indoor localization system using IEEE802.15.4 protocol (FILS15.4)* is integrated. It detects the user locations in indoor environments according to the fingerprints of the target location. The process is divided into the *calibration phase* and the *detection phase* [61,62].

### 9.1. System Architecture

Figure 16 illustrates the overview of FILS15.4 architecture. This system adopts transmitting and receiving devices by Mono Wireless which employs the *IEEE802.15.4* protocol at 2.4 GHz [63]. The transmitter *Twelite 2525* is small with 2.5×2.5 cm and can be powered with a coin battery for a long time. The receiver *Mono Stick* is connected to *Raspberry Pi* over a USB port. To improve the detection accuracy, the sufficient number of receivers should be located at proper locations in the target area.

*Raspberry Pi* receives data from a transmitter, determines the *link quality indication (LQI)* for each transmitter, sends the LQI with the ID to the *MQTT broker* using the *MQTT* protocol. The server receives them from the *MQTT broker*, synchronizes the data from all the receivers, calculates the average LQI with the same transmitter ID, and keeps the results in one record in the *SQLite* database. The previous implementation used a free public MQTT service.

### 9.2. Calibration Phase

The *calibration phase* generates and stores the fingerprint dataset. Each fingerprint consists of *n* LQI values where *n* represents the number of receivers. It represents the typical LQI values when a transmitter is located at the corresponding location (room in *FILS15.4*).

### 9.3. Detection Phase

The *detection phase* detects the current room by calculating the Euclidean distance between the current LQI data and the fingerprint for each room and finding the fingerprint with the smallest distance.

### 9.4. Implementation in Platform

Figure 17 shows the flow of the functions in the platform for integrating this IoT application system. The data synchronization function synchronizes the measured LQI values among all the receivers using the transmitter’s ID, and saves it in the schema data storage. The detection program is implemented as the *plug-in* function in the platform, and receives data through *REST API* services.

We evaluate the integration of *SEMAR* with the *fingerprint-based indoor localization system* by running the system at two floors in the #2 Engineering Building of Okayama University. This system used six receivers to measure LQI from each transmitter. The receiver sent the LQI data every 500 ms to the server through *MQTT* communications. The evaluation results show that *SEMAR* can receive, process, and visualize the data. We also evaluate the data synchronization of the LQI data at the multiple receivers from the same transmitter. Figure 18 shows the synchronized LQI data for *transmitter 1* during 30 s, where LQ*i* for i=1,…,6 indicates the LQI data at *receiver i*. They are saved in the schema data storage and can be accessed from other programs through *REST API*. This system can run without interruptions even if it processes empty LQI data or if error detection occurs. When the system detects an error, it sets the LQI data to the *default value*. According to the evaluation results, the effectiveness of integrating the *SEMAR* system is confirmed.

## 10. Evaluations

In this section, we evaluate the implementation of *SEMAR* IoT server platform.

### 10.1. Performance Analysis

To evaluate the performance of *SEMAR* at the parameter level, first, we investigate the average response time for *MQTT* data communications when the number of IoT devices is increased from 1 to 125. In the experiments, a virtual IoT device is created in the system instead of a real device. Then, each virtual IoT device sends one message through a different topic every second. During this experiment, the CPU usage rate of the machine is also measured.

As the response time, the time difference at a virtual IoT device from the data transmission to the server to the message reception from the server is measured. For *HTTP POST*, it can easily be obtained. When the IoT device sends data to the server, the REST API service returns the response message; however, for *MQTT*, the program is modified to measure the response time where it will send the *MQTT* message to the device when it stores data in the storage.

Figure 19 and Figure 20 show the average response time and the average CPU usage rate when the number of virtual IoT devices is increased from 1 to 125, respectively. The average response time is 315ms and the CPU usage rate is 74% for 125 devices. Thus, *SEMAR* our can handle hundreds of devices with acceptable delay and CPU rate.

### 10.2. The State-of-the-Art Comparative Analysis

We compare the *SEMAR* IoT server platform with 14 recent research works that have the similar approach. In the comparison with the recent related works in the literature, we consider the following features to characterize each proposal:*IoT application*: represents the IoT application that is covered or implemented in each work.*Device management*: indicates the capability of the IoT platform to manage devices (Yes or No).*Communication protocol*: describes the communication protocol utilized in each work.*Data synchronization*: implies the capability to synchronize data across several devices (Yes or No).*Data filtering function*: indicates the implementation of digital filters to process data (Yes or No).*Decision-making assistance*: indicates the implementation of tools to evaluate data or generate alerts based on data obtained (Yes or No).*Flexibility*: shows the abilities to allow to join new devices, to handle different communication settings, to define data types, and to easily interact with external systems (Yes or No).*Interoperability*: represents the ability to be integrated with plural external systems through defined protocols (Yes or No).*Scalability*: demonstrates the capability of processing a number of data simultaneously (Yes or No).

Table 6 compares the fulfillment of the nine features among the 14 related works and the proposed *SEMAR*.

#### 10.2.1. IoT Application

Although the works by Hernández-Rojas et al. in [64], Marcu et al. in [69], and Antunes et al. in [72] have potentials of use in various IoT applications, they have been studied in specific IoT applications. On the other hand, *SEMAR* has been integrated and implemented in several types of IoT applications.

#### 10.2.2. IoT Device Management

All the related works provide functions to add or remove IoT devices. Some works support device management services. Some works include capabilities to define the sensor format for each IoT device dynamically. The work by Trilles et al. in [70] provides the easy-to-use user interface to manage IoT devices. On the other hand, *SEMAR* provides all of the functions on IoT devices.

#### 10.2.3. Communication Protocol

*HTTP* and *MQTT* are the most adopted communication protocols in IoT application platforms. In addition, Del Esposte in [67] and Antunes in [72] introduce *AMQP* as another protocol utilizing TCP connections. Thus, it is suitable for server–client communications [73]. None of the related works reported functions to synchronize data from several devices and digital filters to process sensor data. Only *SEMAR* provides both the data synchronization capability and digital filters to process data.

#### 10.2.4. Decision Making Assistance

For decision-making assistance, a lot of works have offered functions for perspective data analysis based on collected data. The works by Mandava et al. in [18], by Kamienski et al. in [65], by Chiesa et al. in [66], and by Boursianis et al. in [71] applied machine learning algorithms for real-time classifications, and show the results for user interfaces. The work by Hernández-Rojas et al. in [64] utilized message notifications according to a specific data threshold. The work by Trilles et al. in [70] and our *SEMAR* included both of them.

#### 10.2.5. Interoperability and Flexibility

Several works provided interoperability. The works by Hernández-Rojas et al. in [64], by Trilles et al. in [70], and *SEMAR* allow outer programs to process data without changing the existing program in the systems.

Some works consider the flexibility as the IoT application platform. The works by Hernández-Rojas et al. in [64] and by Trilles et al. in [70] provide the capability to dynamically define the sensor format and the data type for each device, similar to *SEMAR*.

However, any work cannot be connected with other *MQTT* servers. Only *SEMAR* flexibly allows users to use other *MQTT* servers, which will allow IoT applications to be easily integrated with *SEMAR*.

## 11. Threats to Validity

There are two kinds of threats to the validity of this research, which are as follows:*Internal validity threat*: validates the potential errors in the *SEMAR* implementation. In this study, *SEMAR* is integrated with five different IoT application systems. Each IoT application utilized various kinds of sensors. Possible threats may occur when submitting invalid or incomplete data. Moreover, the integration of *SEMAR* with the *fingerprint-based indoor localization system* requires the synchronization of data from each receiver to determine the location of the transmitter. To eliminate potential threats, *SEMAR* checks sensor data with the format. In addition, the data synchronization function will provide default values for devices with no data within the data synchronization timeframe.*External validity threat*: validates the generalization ability of the obtained results. We compare the results of *SEMAR* to those of previous IoT-related studies. The primary potential external threat revealed by the comparison results is that not all of the related IoT-related research provided comprehensive and clear explanations of the proposals.

## 12. Conclusions

This paper presented the design and implementation of the *IoT server platform* for integrating various IoT application systems, called *Smart Environmental Monitoring and Analytical in Real-Time (SEMAR)*. It offers *Big Data* environments with *built-in* functions for data aggregations, synchronizations, and classifications with *machine learning*, and *plug-in* functions that access to the data through *REST API*. The platform was implemented and integrated with five IoT application systems. The results confirmed the effectiveness and efficiency of the proposal.

In future studies, we will continue improving the platform by implementing *Rules Engine* and *Complex Event Processing (CEP)* [74] for the data processing. *Rules Engine* will support user-defined rules, actions, and notifications. *CEP* will offer the real-time data analysis based on rule patterns [75]. It will control the device action or deliver messages to users when rule patterns are matched; however, these functions meet issues in parallelism, resource allocations, distributed networks, and multi-rules optimizations [76], which will be studied. Then, we will continue integrating the proposal with various IoT application systems.

## Figures and Tables

**Figure 1 sensors-22-06436-f001:**
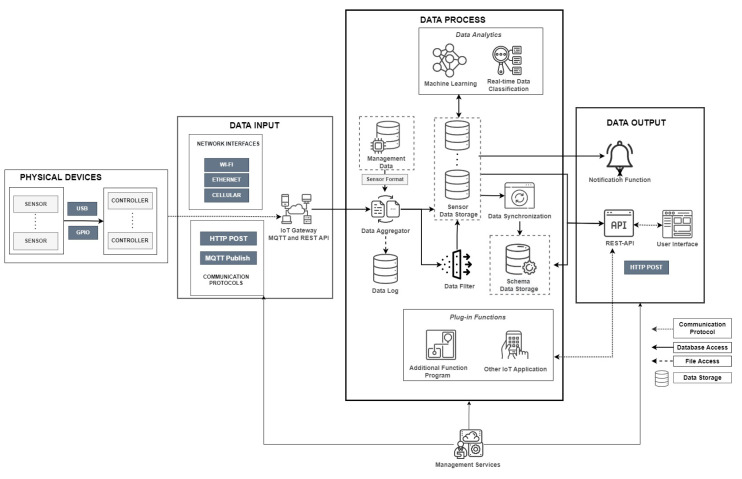
Design overview of *SEMAR* IoT server platform.

**Figure 2 sensors-22-06436-f002:**
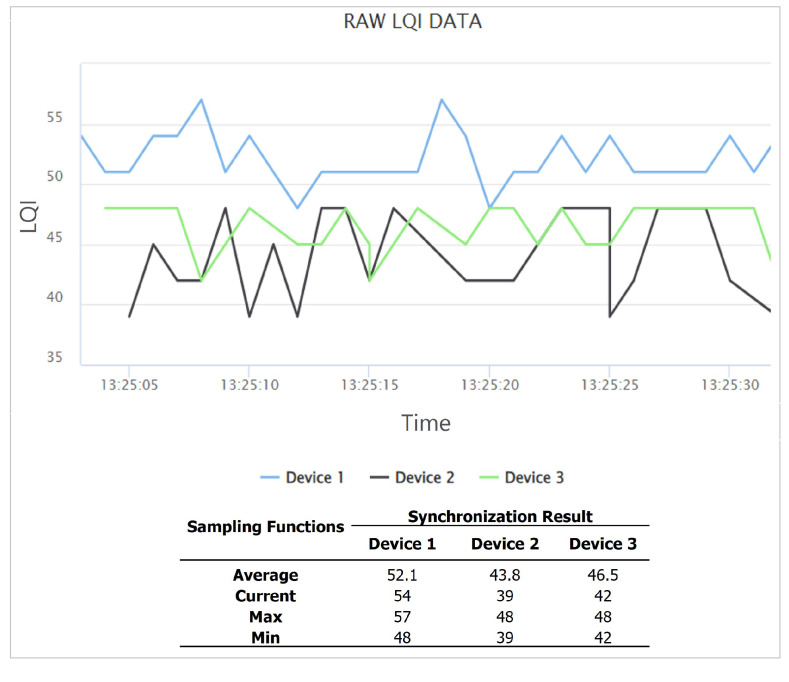
Interface of data synchronization function.

**Figure 3 sensors-22-06436-f003:**
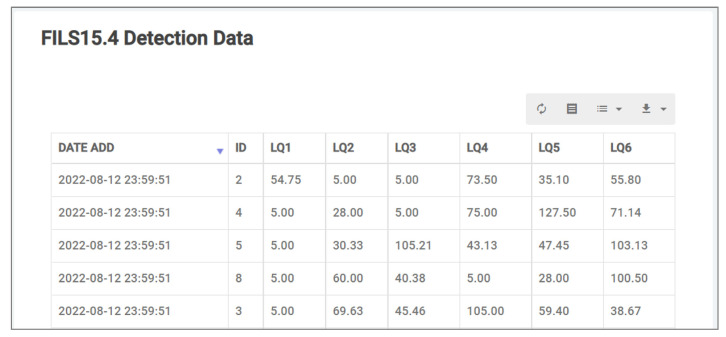
Table of sensor data.

**Figure 4 sensors-22-06436-f004:**
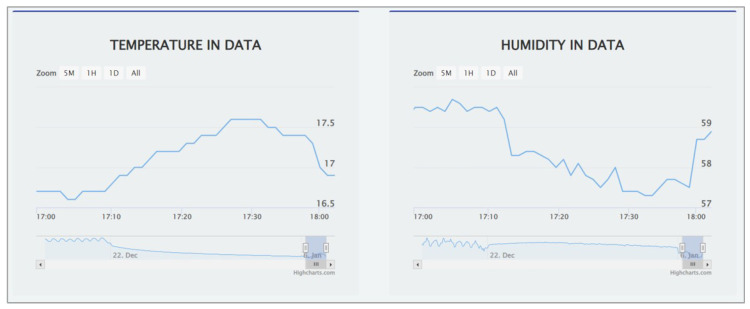
Graphs of sensor data.

**Figure 5 sensors-22-06436-f005:**
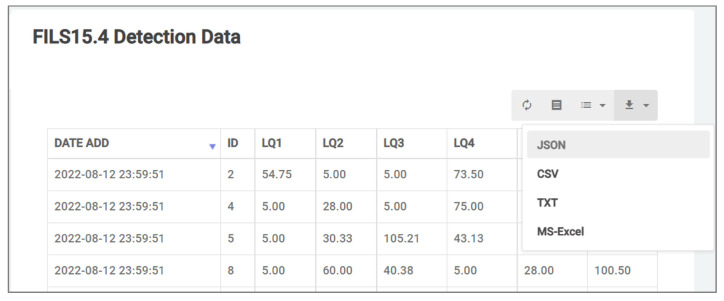
Data export interface.

**Figure 6 sensors-22-06436-f006:**
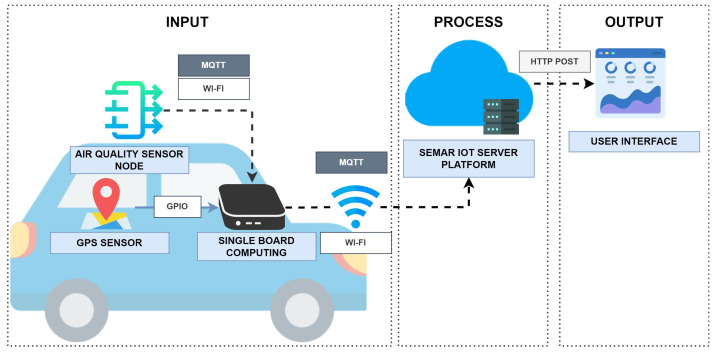
System overview of *air quality monitoring system*.

**Figure 7 sensors-22-06436-f007:**
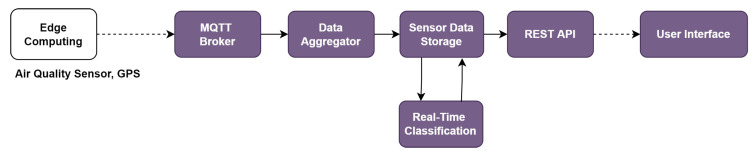
Function flow for *air quality monitoring system* in platform.

**Figure 8 sensors-22-06436-f008:**
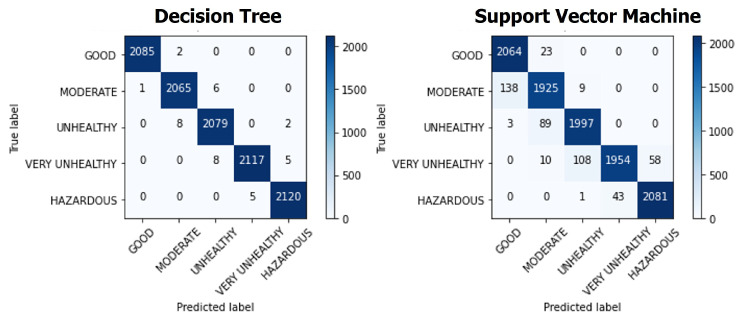
Confusion matrices of *Decision Tree* and *Support Vector Machine*.

**Figure 9 sensors-22-06436-f009:**
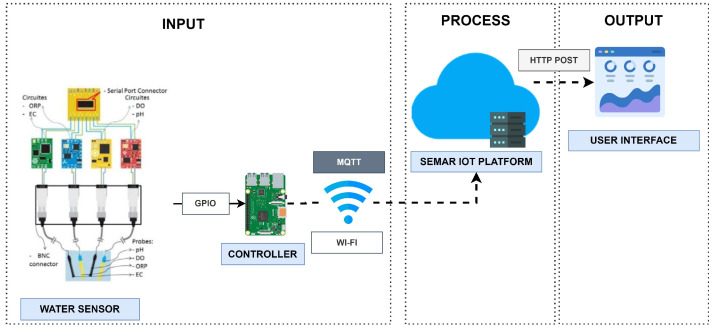
The system overview of the *water monitoring system*.

**Figure 10 sensors-22-06436-f010:**
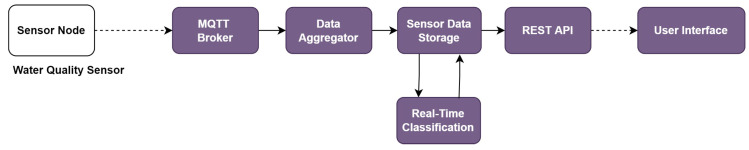
Function flow for *water quality monitoring system* in platform.

**Figure 11 sensors-22-06436-f011:**
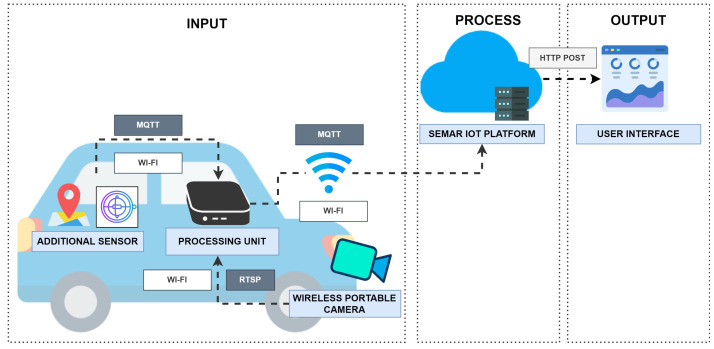
System overview of *road condition monitoring system*.

**Figure 12 sensors-22-06436-f012:**
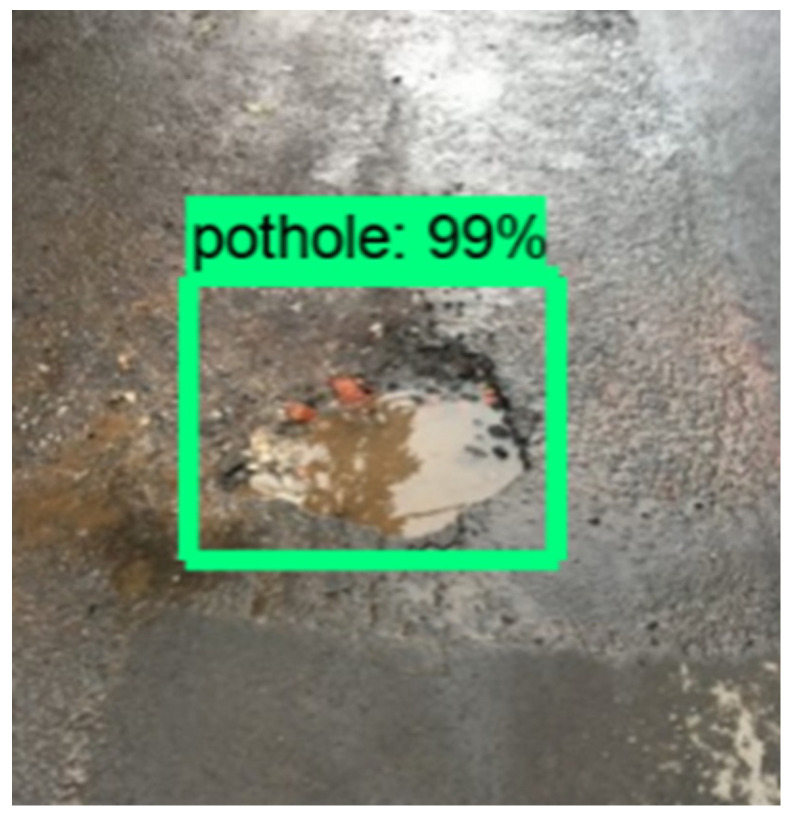
Detected pothole example.

**Figure 13 sensors-22-06436-f013:**
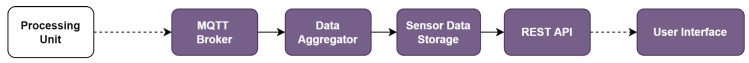
Function flow for *road condition detection system* in platform.

**Figure 14 sensors-22-06436-f014:**
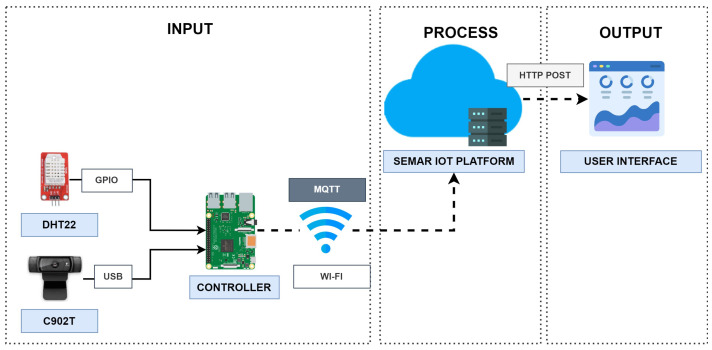
System overview of *AC-Guide*.

**Figure 15 sensors-22-06436-f015:**
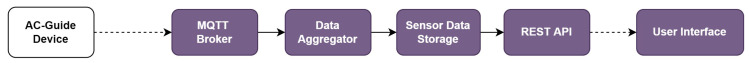
Function flow for *AC-Guide* in platform.

**Figure 16 sensors-22-06436-f016:**
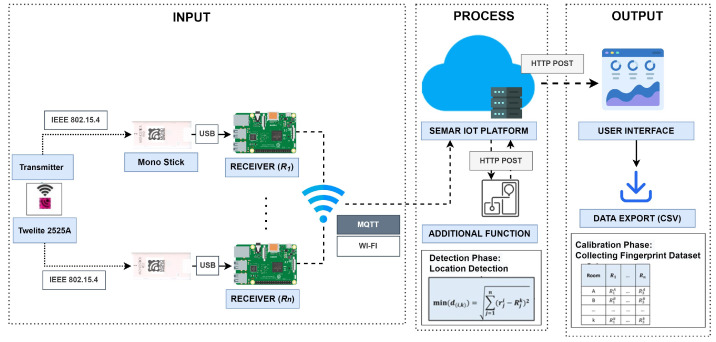
System overview of *FILS15.4*.

**Figure 17 sensors-22-06436-f017:**
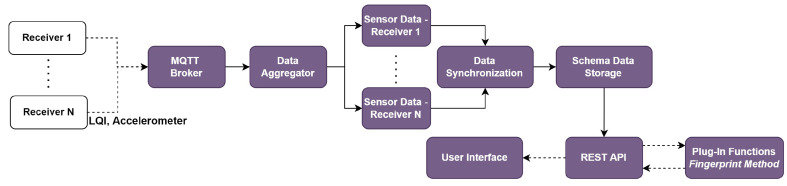
Function flow for *FILS15.4* in platform.

**Figure 18 sensors-22-06436-f018:**
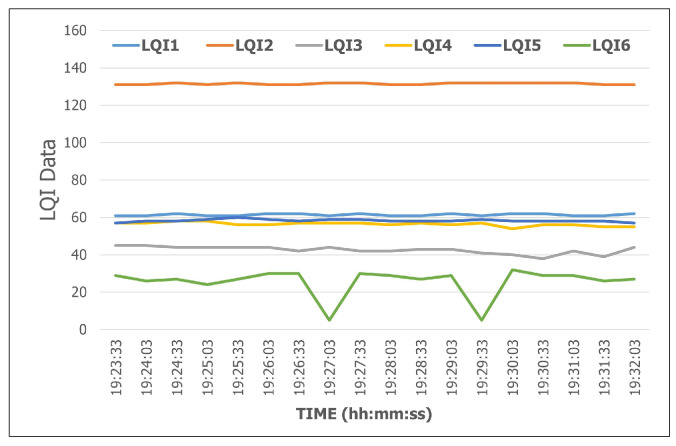
LQI data of *transmitter 1*.

**Figure 19 sensors-22-06436-f019:**
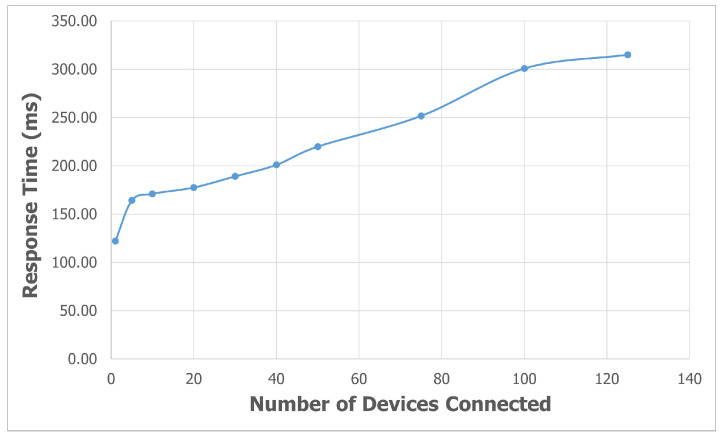
Average response time for *MQTT* communications with different numbers of devices.

**Figure 20 sensors-22-06436-f020:**
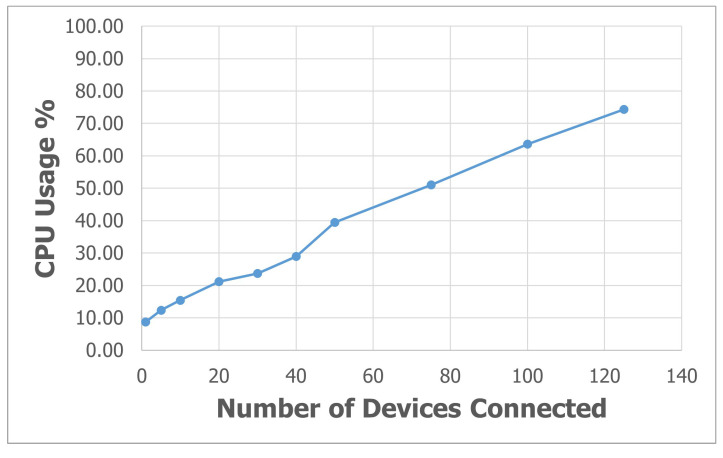
Average CPU usage rate with different numbers of devices.

**Table 1 sensors-22-06436-t001:** Nomenclature used in the paper.

Parameters	Description
	*Decision Tree* algorithm:
*t*	the node of decision tree
*n*	number of targets classes
P(i|t)	the probability of the specific data class *i* in node *t*
	*Support Vector Machine* algorithm:
$y_{i}$	the class label of dataset
$\alpha_{i}$	the learned weight
$x_{i}$	the support vector
*x*	the labeled training sample data.
K()	the kernel function
	*Radial Basis Function* kernel:
*l*	the length scale of the kernel RBF
$d(x_{i},x_{j})$	the Euclidean distance between $x_{i}$ and $x_{j}$

**Table 2 sensors-22-06436-t002:** Technology specifications for implementation of *SEMAR* IoT server platform.

IoT Model	Function	Component	Description
Input	MQTT	MQTT Broker MQTT Supports	Mosquitto v2.0.10 MQTT v5.0, v3.1.1, and v3.1
	REST API	Libraries and Framework Communication Supports	Tornado Web Server, PyMongo, JSON HTTP-POST
	Network Interfaces	Network Interfaces Supports	Wi-Fi, Ethernet, Cellular
Process	Server	Operating System Memory	Ubuntu 18.04.5 LTS 6Gb
	Data Storage	Services	MongoDB v3.6.3
	Data Aggregator	Libraries and Framework Communication Supports	Tornado Web Server, PyMongo, JSON, Paho HTTP-POST and MQTT
	Data Filter	Libraries and Framework	PyMongo, JSON, Numpy, Scipy and KalmanFilter
	Data Synchronization	Libraries and Framework	PyMongo, JSON, Pandas, Statistics and Threading
	Machine Learning and Real-time Data Classification	Libraries and Framework	sklearn, Pandas, PyMongo, JSON, and Threading
Output	User Interfaces and Data Export	Programming Language Libraries and Framework Web services Development Pattern Supported browsers	PHP, CSS, HTML and Javascript CodeIgniter, Bootstrap, JQuery, HighChart JS, DataTables, OpenStreetMap Apache v2.4.29, PHP 7.2.24 MVC Google Chrome, Firefox, Opera
	REST API	Libraries and Framework Communication Supports	Tornado Web Server, PyMongo, and JSON HTTP-POST
	Notification Functions	Libraries and Framework Notification supports Email Service	PyMongo, JSON, Paho, smtplib Email and MQTT Postfix
Management	Management Services	Libraries and Framework Communication Supports	Tornado Web Server, PyMongo and JSON HTTP-POST

**Table 3 sensors-22-06436-t003:** Evaluation of air quality monitoring classification model.

Features	Algorithm	Mislabel	Accuracy	MSE
Air Quality	Support Vector Machine	605/10,053	0.94239	0.05761
	Decision Tree	43/10,053	0.99591	0.00409

**Table 4 sensors-22-06436-t004:** Experiment setup for hyper parameter optimizations in air quality monitoring.

Component	Specification
Operating System	Windows 10 Enterprise, 64-bit
Processor	AMD Ryzen 5 3550H
RAM memory	8.0 GB
Machine Learning Library	Scikit-learn [55]
Machine Learning Method	*Support Vector Machine* and *Decision Tree*
Datasets	25,000 rows air quality data (5 labels, 5 features)

**Table 5 sensors-22-06436-t005:** Evaluation of water quality monitoring classification model.

Features	Algorithm	Mislabel	Accuracy	MSE
Water Quality	Support Vector Machine Decision Tree	289/45,397 34/45,397	0.9936 0.9993	0.0064 0.0007

**Table 6 sensors-22-06436-t006:** State-of-the-art comparison between the existing related studies and the proposed solution.

Work Reference	IoT Application	Device Management	Data Synchronization	Data Filter	Decision-making assistance	Flexibility	Interoperability	Scalability	Communication Protocol
[17]	Indoor Air Quality	✓	✗	✗	✗	✓	✗	✓	HTTP
[64]	Smart Agriculture	✓	✗	✗	✓	✓	✓	✓	MQTT
[18]	Air Pollution	✓	✗	✗	✓	✗	✗	✓	HTTP
[19]	Water Management	✓	✗	✗	✗	✗	✗	✓	HTTP
[65]	Water Management	✓	✗	✗	✓	✓	✓	✓	MQTT
[21]	Air Pollution	✓	✗	✗	✗	✓	✓	✓	MQTT
[66]	Indoor Air Quality	✓	✗	✗	✓	✗	✓	✓	MQTT
[67]	Smart City	✓	✗	✗	✗	✗	✗	✓	HTTP & AMQP
[68]	Smart Industry	✓	✗	✗	✓	✓	✓	✓	MQTT
[69]	Smart Agriculture and Smart City	✓	✗	✗	✗	✓	✓	✓	MQTT
[70]	Smart Farming	✓	✗	✗	✓	✓	✓	✓	MQTT
[22]	Smart Building	✓	✗	✗	✓	✗	✓	✓	HTTP & Web Socket
[71]	Smart Irrigation	✓	✗	✗	✓	✗	✗	✓	MQTT
[72]	Smart Green and Smart City	✓	✗	✗	✗	✓	✓	✓	HTTP, MQTT, AMQP
*SEMAR*	Various IoT applications	✓	✓	✓	✓	✓	✓	✓	HTTP & MQTT

## Data Availability

Not applicable.

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
