# Peer review of "Design and Implementation of SEMAR IoT Server Platform with Applications"

_sensors, 2022, doi:10.3390/s22176436_

Round 1

Reviewer 1 Report

The authors proposed a design of an IoT server platform. Each component is described with a few paragraphs. Here are my questions and comments:

1.  I think the purpose of Figures 2-5 are for showing the ability and UI of the implemented API/system. Does the API or the source code or whatever publicly/commercially available for the others to deploy such a system?

2. I can find "REST-API", "REST API" and "Rest API" throughout the text. I think it is better to be consistent.

3. The related works section only includes 8 references. As IoT and smart cities are huge research fields, I suggest the authors to search for more related works.

4. Page 8, under Eq. (3). "d(x_i, x_j" -> "d(x_i, x_j)"

5. Page 2. above Section 2. "Section 5,6,7,8 and 9" -> "Sections 5, 6, 7, 8 and 9" or "Sections 5-9"

Author Response

Dear Reviewer,
We appreciate your time in reviewing our paper and providing valuable comments. Your insightful and valuable comments led to possible improvements in the current version. The authors have carefully considered the comments and tried our best to address every one of them. We hope that the revised manuscript meets your high standards.

We provide the point-by-point responses in our response letter.
Please see the attached file.
All modifications in the manuscript have been highlighted in yellow.

Reviewer 2 Report

The manuscript address a timely problem and is written well and should be accepted after this revision. However, authors need to spend time to address each and every comment carefully. Also they are advised to highlight the changes in the revised manuscript. Especially, abstract needs to be re-written to highlight novelty of the proposed approach. Related work needs to be strengthened. Below are my few observations:
1. The English language used in the article has to be polished.
2. There have been numerous works recently proposed. How is this article different from them?
3.  Authors should discuss these related works to enhance the paper work such as: DLTIF: Deep Learning-Driven Cyber Threat Intelligence Modeling and Identification Framework in IoT-Enabled Maritime Transportation Systems; BDEdge: Blockchain and Deep-Learning for Secure Edge-Envisioned Green CAVs; Design of anomaly-based intrusion detection system using fog computing for IoT network. P2IDF: A privacy-preserving based intrusion detection framework for software defined Internet of Things-fog (SDIoT-Fog); PEFL: Deep Privacy-Encoding-Based Federated Learning Framework for Smart Agriculture; Toward design of an intelligent cyber attack detection system using hybrid feature reduced approach for iot networks; A distributed ensemble design based intrusion detection system using fog computing to protect the internet of things networks; TP2SF: A Trustworthy Privacy-Preserving Secured Framework for sustainable smart cities by leveraging blockchain and machine learning; An ensemble learning and fog-cloud architecture-driven cyber-attack detection framework for IoMT networks.
4. How are the hyper parameters used in the work chosen? Is it random or did the authors used any parameter tuning method?
5. A more detailed analysis on the results obtained has to be added.
6. What are the threats to validity of the proposed model?
7. How can the proposed work be enhanced in real-time network scenario?

Author Response

(The authors gave the same response as above.)

Reviewer 3 Report

In the article authors present the design and implementation of the IoT server platform called Smart Environmental Monitoring and Analytical in Real-Time (SEMAR) for integrating IoT application systems using standards. SEMAR offers Big Data environments with built-in functions for data aggregations, synchronizations, and classifications with machine learning.

The IoT implementation examples presented in Chapters 5-9 are interesting. However, the description of the SEMAR platform itself requires editing and improvement. Currently, the article consists of many, very short subsections - some of them are one-sentence. It's not like a scientific article, but more like a dictionary or encyclopedia. Much of the description that is generally known can be moved to the Nomenclature, which should be added at the beginning of the article. Then, the description of the SEMAR platform and its basic functionalities and properties should be presented in compact paragraphs - not fragmented sub-chapters.

In my opinion, the current form of the article is unacceptable in reputable publishing houses and should not be allowed for publication. The tendency to simplify articles by means of "shredding" the article into a large number of subsections - not necessarily showing objectively a large number of issues raised in articles is observed more and more often.

Author Response

(The authors gave the same response as above.)

Reviewer 4 Report

The paper is interesting, proposing a generic IoT platform having AI-based classification capabilities. The positive aspect of the paper is given not only due this generic platform, but also the capacity to use it in order to implement several monitoring applications, apparently very different one to another.

The state-of-the-art is very briefly described. It may be improved, due to the diversity of publications in the area.

The platform is very briefly described, without mentioning any performance at parameter level. Also, during each application implementation, it is analyzed only the capacity to implement these specific applications, but nothing about the performances of the specific application or about the efficiency of the proposed system, at least in comparison to a dedicated platform for each type of application. 

Author Response

(The authors gave the same response as above.)

Round 2

Reviewer 1 Report

I am fine with the technical details. Here are some minor suggestions regarding the English.

1. Usually when we refer to more than one section/equation/figure/etc., we should use plural form. That is, it should be "Sections 5, 6, 7, 8 and 9" instead of "Section 5, 6, 7, 8 and 9". This is a minor issue though.

2. The term REST appears in Section 1 for the first time, but without quoting the full name of it. (It is quoted in Section 2, which is the second time this term appears). I suggest to also give the full name of REST when it appears at the first time.

3. I can see many "does" in the updated manuscript. Somehow the English is a little bit strange for me. For example, "The undefined data does a data ...", which I think the authors want to refer "does" to the word "represents" in the previous sentence. I think the authors may change most "does" throughout the text into suitable verbs depend on the context, such as "represents", "is", "denotes", etc.

Author Response

Dear Reviewer,
We again appreciate your kindness in reviewing our paper and providing valuable comments. Your insightful and valuable comments help improve the quality of the current version manuscript. The authors have carefully considered the comments and tried our best to address every one of them. We hope that the revised manuscript meets your high standards.

We provide the point-by-point responses in our response letter.
Please see the attached file.
All modifications in the manuscript have been written in blue.

Reviewer 2 Report

Addressed all my previous comments. Accept the paper in current form.

Author Response

Dear Reviewer,
We again appreciate your kindness in reviewing our paper and providing valuable comments. Your insightful and valuable comments help improve the quality of the current version manuscript. All authors have reviewed and agreed to the submission of the revised manuscript. We hope that the revised manuscript meets your high standards.

We provide the point-by-point responses in our response letter.
Please see the attached file.
All modifications in the manuscript have been written in blue.

Reviewer 3 Report

All my suggestion has been taken into account and indicated problems have been solved. Additional comments, which have been added on a request of the other reviewers also improve the scientific level of the paper. Therefore, in my opinion, the paper in the current version is suggested to be accepted in Sensors Journal.

Author Response

(The authors gave the same response as above.)

Reviewer 4 Report

I highly appreciate the authors' effort to increase the quality of the paper both in terms of content and of presentation quality.

Author Response

(The authors gave the same response as above.)
